# Rapid Multi-Hybridisation FISH Screening for Balanced Porcine Reciprocal Translocations Suggests a Much Higher Abnormality Rate Than Previously Appreciated

**DOI:** 10.3390/cells10020250

**Published:** 2021-01-28

**Authors:** Rebecca E O’Connor, Lucas G Kiazim, Claudia C Rathje, Rebecca L Jennings, Darren K Griffin

**Affiliations:** School of Biosciences, University of Kent, Canterbury Kent CT2 7NJ, UK; rebeckyoc@gmail.com (R.E.O.); lkiazim@gmail.com (L.G.K.); C.C.Rathje@kent.ac.uk (C.C.R.); beccajennings3@gmail.com (R.L.J.)

**Keywords:** artificial insemination, fertility, FISH, pig, hypoprolificacy, reciprocal translocation

## Abstract

With demand rising, pigs are the world’s leading source of meat protein; however significant economic loss and environmental damage can be incurred if boars used for artificial insemination (AI) are hypoprolific (sub-fertile). Growing evidence suggests that semen analysis is an unreliable tool for diagnosing hypoprolificacy, with litter size and farrowing rate being more applicable. Once such data are available, however, any affected boar will have been in service for some time, with significant financial and environmental losses incurred. Reciprocal translocations (RTs) are the leading cause of porcine hypoprolificacy, reportedly present in 0.47% of AI boars. Traditional standard karyotyping, however, relies on animal specific expertise and does not detect more subtle (cryptic) translocations. Previously, we reported development of a multiple hybridisation fluorescence in situ hybridisation (FISH) strategy; here, we report on its use in 1641 AI boars. A total of 15 different RTs were identified in 69 boars, with four further animals XX/XY chimeric. Therefore, 4.5% had a chromosome abnormality (4.2% with an RT), a 0.88% incidence. Revisiting cases with both karyotype and FISH information, we reanalysed captured images, asking whether the translocation was detectable by karyotyping alone. The results suggest that chromosome translocations in boars may be significantly under-reported, thereby highlighting the need for pre-emptive screening by this method before a boar enters a breeding programme.

## 1. Introduction

The domestic pig (*Sus scrofa*) provides 30–40% of the meat consumed worldwide, making it the leading source of meat protein globally (United Nations Food and Agriculture Board 2020). Estimates by the United Nations suggest that global pork production will reach 115.6 million tonnes in 2029 [1]. The goal for pig breeding companies is therefore to support this demand while increasing profitability, reducing wastage and environmental damage and, simultaneously, ensuring that the highest levels of animal welfare are maintained.

The use of artificial insemination (AI) in pig breeding is relatively recent compared to its use in cattle. Commercial application of porcine AI began in the 1980s and has grown exponentially since [2]. Fertility assessment in boars awaiting AI service is commonly measured through semen analysis, examining volume and concentration of the sperm-rich fraction, progressive motility, morphological abnormalities and reacted acrosomes [3]. Computer Assisted Semen Analysis (CASA), the most commonly used tool, is very objective [4] and it is known that its parameters correlate to fertility [5]. However, even using CASA, only the extreme outliers can be detected and the ability to detect variation in fertility is limited [6,7]. Farrowing rates and litter size are also commonly examined following AI since a decrease in litter size is considered the primary indicator of boars displaying sub-optimal fertility. Sometimes the decrease is not so evident, however, as it takes many deliveries to observe a significant variation. In these cases the economic damage will have already been done as many fertilizations will have been done in the meantime. For this reason, preventive screening of any factor that causes a reduction in litter size is imperative. The gestation period in swine is 115 days and, on average, a sow will farrow 12 piglets per pregnancy [8]. Therefore, fertility problems take some time to identify in any given boar and the boar may be towards the middle to end of its reproductive life before robust data is gathered and the boar taken out of service. By this time, any underlying genetic abnormality would have been passed on to a large number of offspring, thereby perpetuating the problem. There are also environmental considerations here. An on-going pregnancy, in addition to the 115-day gestation, comprises a 26-day lactation and 5-day weaning service interval. Such a cycle has a similar environmental footprint, regardless of the litter size; thus, to go through the whole process for only half the number of piglets is a significant “cost” for a low return, both financially and environmentally. In these terms, it would be better if the sow had not become pregnant at all and returned to service in 21 days. Identifying fertility issues that may result in compromised fertility parameters such as reduced litter size *before an animal is entered into the AI breeding program* is therefore vital.

Chromosomal abnormalities (mostly balanced reciprocal translocations—RTs) are reportedly seen in as many as 0.47% of AI boars awaiting service [9]. More than 130 different RTs have been identified to date. Balanced RTs can be inherited from a parent or occur *de novo* during meiosis. Carriers of balanced RTs cause a reduction in farrowing rate litter size because they can impede meiosis and produce up to 50% unbalanced gametes [9,10,11,12]. Offspring from unbalanced gametes carry partial trisomy and monosomy of the translocated segments and result in early embryo loss or malformed piglets [10,11,12]. Around half of all boars exhibiting hypoprolificacy are reciprocal translocation carriers, despite displaying a normal phenotype and semen parameters [12]. The ultimate outcome of using a hypoprolific boar is embryonic loss, resulting in litters that are 25–50% smaller [13,14]. Porcine cytogenetics peaked during the latter part of the 20th century with several continental European cytogenetic programmes, the largest based at the National Veterinary School of Toulouse, France [15]. More recently, however, the number of labs has decreased, despite the clear association between hypoprolificacy and chromosomal aberrations. This risks perpetuation of the problem as RTs may be passed on to the offspring. Current translocation screening by these centres is performed by G-banding and routine karyotyping which, while simple and cost effective, requires specialist knowledge of the pig karyotype. Many translocations (even those of large size) are difficult to identify by banding alone (this depends on the portions of chromosomes that are exchanged, what banding profile they have and the quality of the preparation), while some (cryptic translocations) cannot be detected by banding, regardless of preparation quality.

We recently reported the development of a novel screening protocol using fluorescence in situ hybridisation (FISH) based on multiple hybridisations of sub-telomeric probes [11]. The ability to detect cryptic translocations was evident and a single 5:6 reciprocal translocation that would not have been detected by classical means was demonstrated. Here we report on the widespread use of this approach through the establishment of a cytogenetic screening service used to screen, to date, over 1600 animals. Our efforts allow the reappraisal of the question of the incidence of chromosome abnormalities, particularly those missed by traditional (G-banding) methods.

## 2. Materials and Methods

Blood samples intended for the screening of boars were provided in heparin tubes from a range of European pig breeding companies. Samples were collected as part of standard procedures used for commercial evaluation by in-house trained veterinarians via standard phlebotomy. Whole blood samples were cultured for 72 h in PB MAX Karyotyping medium (Gibco, Thermo Fisher Scientific, Waltham, MA, USA) at 37 °C, 5% CO_2_. Cell division was arrested by the addition of colcemid at a concentration of 10.0 μg/mL (Gibco, Thermo Fisher Scientific, Waltham, MA, USA) for 35 min prior to hypotonic treatment with 75 M KCl (12 min) and fixation using freshly prepared 3:1 methanol:acetic acid (three changes). Immobilization to glass slides preceded banding and FISH analysis. For pseudo-G-banding analysis slides were stained with VECTASHIELD^®^ (Vector Labs) antifade medium containing DAPI counterstain to visualise the chromosomes. Image capturing was performed using an Olympus BX61 epifluorescence microscope with a cooled CCD camera and SmartCapture (Digital Scientific UK) system. SMARTTYPE software (Digital Scientific UK) was used for karyotyping purposes after being custom-adapted for porcine karyotyping according to the standard karyotype as established by the Committee for the Standardized Karyotype of the Domestic Pig [16]. FISH was performed following O’Connor et al. [11]. Briefly, a bespoke device developed in-house, based on the CytoCell Multiprobe system (Oxford Gene Technologies—OGT), allowed multiple individual hybridizations (one for each chromosome) on each slide. Using sub-telomeric probes (one in Texas Red and one in FITC)—fluorescein isothiocyanate at the proximal and distal ends of each chromosome permits the detection of any RT, however cryptic. Probes were dried onto the “coverslip” component of the CytoCell Multiprobe system, in the top left-hand corner: first chromosome 1, then chromosome 2, and so on. At the same time, a 1 µL drop of chromosome suspension was dried onto each of the corresponding squares of the “slide” part of the device, then dehydrated through an ethanol series (2 min each in 2× SSC, 70%, 85% and 100% ethanol). Re-hydration of the probes in 1 µL of HybI (OGT) preceded “sandwiching” of the slide and coverslip together. Probe and target DNA were subsequently denatured on a 75 °C hotplate for 5 min prior to overnight hybridization at 37 °C in a moist chamber. Post-hybridization washes consisted of 2 min in 0.4× SSC at 72 °C and 30 s in 2× SSC/0.05% Tween 20 at room temperature. Slides were then counterstained using DAPI in VECTASHIELD^®^ anti-fade medium (Vector Labs).

Each chromosome was analysed for the presence of a translocation (i.e., the absence of a translocation was indicated by signals (red and green, top and bottom) on the same chromosome). Analysis of 50 cells for the presence of extra X chromosomes allowed for the detection of XX/XY chimerism. Each boar was identified for each chromosome in this regard. The presence of an RT was confirmed independently by a misplaced signal on both the chromosomes involved in the translocation.

## 3. Results

In total, 1641 boars were screened over an eight-year period. Of these, 1116 were screened using the Multiprobe FISH method (Figure 1) described above and 525 by karyotyping and, if an abnormality was found, confirmed by FISH. In addition, we introduced screening for XX/XY chimerism (analysing 50 cells for the presence of extra X chromosomes—see above) approximately halfway through the process.

Results are summarised in Table 1. During this time, 15 different reciprocal translocations were identified in a total of 69 animals, with an additional four animals identified as XX/XY chimeric. Therefore, of all animals screened, 4.5% had a chromosome abnormality and 4.2% had an RT. We calculated an incidence of 0.88% (95% confidence intervals 1.33–0.43%), given that some RTs were clearly from related boars (Table 1).

We then revisited each of the translocations detected by FISH and, using karyotype analysis, analysed each sample to assess whether the translocation would have been identifiable using karyotyping alone. We answered “yes” for 10 of the 15 (e.g., Figure 2) and “no” for 5 of the 15 (e.g., Figure 3).

## 4. Discussion

The results presented here demonstrate that chromosomal translocations persist in the pig breeding population despite widespread eradication programmes over many years. Our opinion therefore is that it is vital that routine screening is continued. The published incidence of RTs is 0.47% [9]; however, we found a value of nearly twice that (0.88%), although our confidence interval was lower (0.43%). Given the perils of ascertainment bias, this value alone is not enough to arrive at the conclusion that chromosome abnormalities have previously been underscored. Nevertheless, we re-visited the karyotypes that we revealed as having RTs and found, through FISH analysis, that around one third of these would not have been detected using karyotyping alone. Given that we did this in the knowledge that each of these samples did indeed have an abnormality (and we knew exactly which chromosomes were involved), this is almost certainly an underestimate and it is possible that more would have been missed.

Proof of principle for the technology used in this study had previously been established when we detected that a single boar carrying a cryptic RT t(5:6) was diagnosed as normal using standard karyotyping [11]. This boar was included in this data set; however, this study takes this work a step further, establishing that further translocations undetectable by karyotyping alone were detected in further animals. Taken together, a total of 73 boars out of the 1641 analysed had a translocation or XX/XY chimerism. The four cases of XX/XY chimerism were unrelated to one another; however, the identification of repeated translocations often initiated the screening of their progeny and family population. Moreover, while our technology did not differentiate balanced from unbalanced RTs, any net gain or loss of DNA through an unbalanced translocation would not lead to a liveborn boar, or would lead to one with significant congenital abnormalities, and thus would not be considered in our analysis.

Re-analysis of previously identified translocations using karyotyping effectively suggests that by using our Multiprobe FISH-based screening method, 50% more balanced RTs could be detected. The additional value of being able to screen for XX/XY chimerism means that these results highlight the higher resolution of FISH screening for AI boars. In other words, the previously published incidence of 0.47% is almost certainly an underestimate and should be reappraised in the light of these results.

Pork production is a growing market globally and plays an important role as the demand increases for sustainable and economically accessible protein [17]. Pork is currently the most consumed meat globally and encompasses over thirty percent of all meat consumed [1]. This demand for pork meat is increasing and is expected to rise from the current 106 metric kilotons of carcass weight consumed globally to 127 metric kilotons by the year 2029 [18]. The pork production industry itself has a wide range of business stakeholders, from major corporate players such as Smithfield Foods, Tyson Foods Inc., Danish Crown and Wan Chau International Ltd. all the way down to basic individual small farms [19]. In 2019, the pork industry world was worth over 396 billion US dollars and it is projected to reach over 500 billion by the year 2027 [20]. The ultimate goal for most of these businesses will be to not only meet the demand for pork but to maximise their profits [21]. As this demand increases and along with the use of new technologies, such as the widespread use of AI, biological issues that can reduce productivity and profit inevitably become more significant. This is most notably seen in the current African swine fever epidemic that has cost China and Europe billions and which, if it arrived in the United States, could potentially cost the US over fifty billion US dollars [22]. For this reason, regular cytogenetic screening by state-of-the-art methods, such as those reported in this paper, is increasingly becoming a priority.

## 5. Conclusions

The technology described here was proven to be a reliable and effective method to screen for translocations that are undetectable through traditional karyotyping. Due to the financial and environmental damage RTs can have on food production, we propose that any abnormality should be identified quickly and accurately by the approached described herein, rather than standard karyotyping.

## Figures and Tables

**Figure 1 cells-10-00250-f001:**
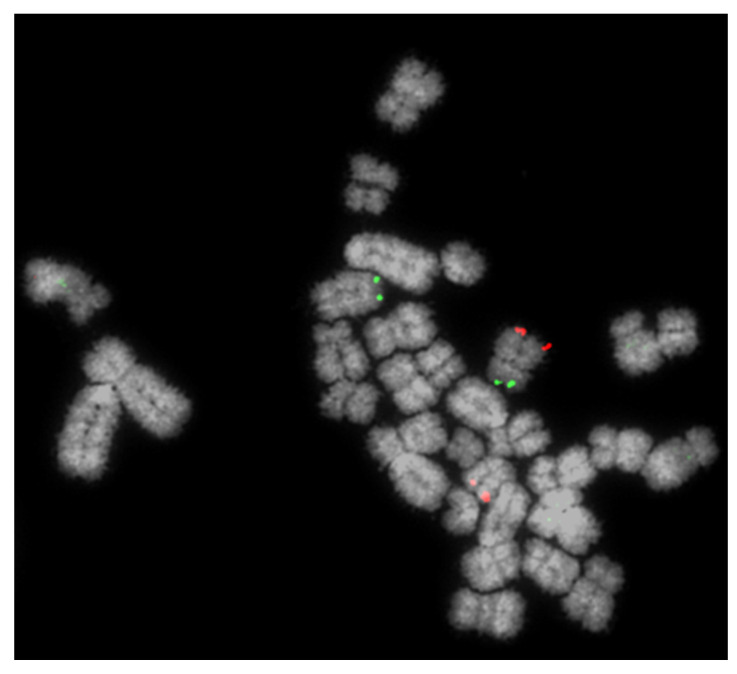
Detection of a reciprocal translocation of chromosomes 4 and 5 using the Multiprobe fluorescence in situ hybridisation (FISH) approach. BACs (bacterial artificial chromosomes) for the short (p) arm (green) and long (q) arm of chromosome 4 are clearly visible on the same (normal) chromosome but on different chromosomes where the RT is involved.

**Figure 2 cells-10-00250-f002:**
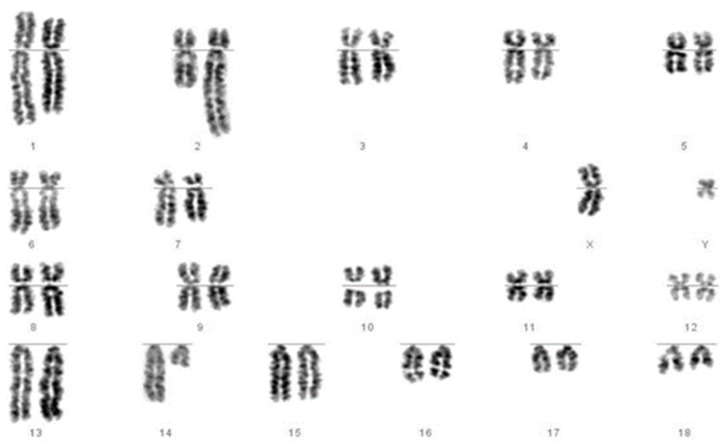
Chromosomes of a boar identified as carrying a reciprocal translocation of chromosomes 2 and 14. The longer chromosome 2 and shorter chromosome 14 are clearly visible.

**Figure 3 cells-10-00250-f003:**
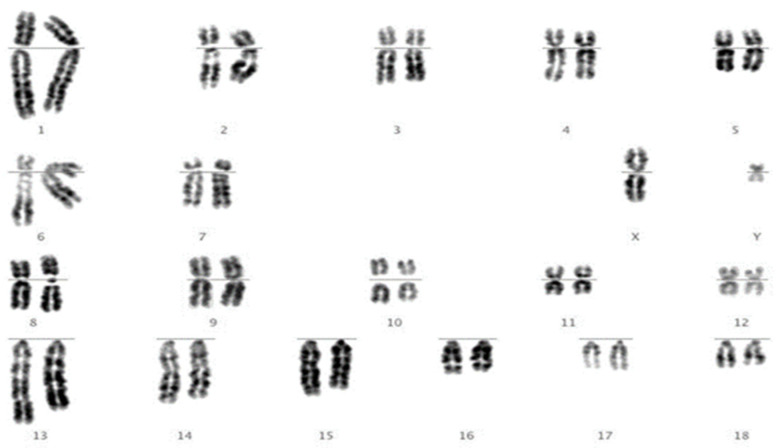
Traditional banded metaphase chromosomes of a boar identified as carrying a reciprocal translocation of chromosomes 9 and 12. After analysis of ten metaphases, even with prior knowledge of the translocation from the FISH result, we were confident that we would not have identified this translocation by karyotyping alone.

**Table 1 cells-10-00250-t001:** Summary of porcine screening results from 2016–2019 using the multiprobe FISH method and standard karyotyping.

Chromosome Abnormality	Number of Cases	Identifiable with Karyotyping?
t(5;6)	1	No
t(3;9)	2	Yes
t(7;10)	20	Yes
t(3;7)	1	Yes
t(16;17)	1	Yes
t(9;13)	1	Yes
t(3;9)	1	Yes
t(1;13)	1	No
t(1;17)	1	Yes
t(2;14)	1	Yes (Figure 2)
t(4;5)	14	No
t(7;10)	1	Yes
t(9;10)	1	No
t(9;12)	1	No (Figure 3)
t(10;15)	1	Yes
XX/XY chimeric	4	N/A

## Data Availability

All data is contained within the manuscript; individual FISH images and karyotypes available from the authors on request.

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
