# Peer review of "Rapid Multi-Hybridisation FISH Screening for Balanced Porcine Reciprocal Translocations Suggests a Much Higher Abnormality Rate Than Previously Appreciated"

_cells, 2021, doi:10.3390/cells10020250_

Round 1
Reviewer 1 Report
Dear Authors,
I have read your article with interest and I consider it suitable for publication. I only ask you to read the few points I have made that I believe deserve attention.
Line 43: I agree with this statement, but in my opinion it must also be considered that sometimes the decrease is not so evident and it takes many deliveries to observe a significant variation. In this case, however, the economic damage has already been done as many fertilizations have been done in the meantime. For this reason, preventive screening is useful (This observation is partly explained in the following lines).
Line 45: The value refers to 2014: is there a more recent data?
Line 60: Would it be possible to insert a reference?
Line 75: In my opinion the identification of translocations involving much larger fragments is however difficult. It depends on what portions of chromosomes are exchanged and what banding profile they have. I would prefer that all references to dimensions be eliminated.
Line 106: I suggest to add two lines regarding the G band protocol for karyotypes (considering that there are two figures in the paper)
Line 111: What does this “system” consist of?
Line 122: I would also indicate the 95% confidence interval of this percentage (1.33-0.42). In my opinion it is interesting to note that the value reported on line 59 is included in this range.
Line 170: I believe that using the described techniques it is not possible to establish whether a translocation is balanced or not. Only the use of other molecular approaches can give this result (as demonstrated in humans: Patsalis et al., Eur J Hum Genet 12:647--653 (2004); Gribble et al., J Med Genet 42:8--16 (2005); De Gregori et al., J Med Genet 44:750--762 (2007); Sismani et al., Mol Cytogenet 1:1--15 (2008).
Author Response
We are grateful to reviewer 1 for their insightful comments. They are addressed point by point below.
I have read your article with interest and I consider it suitable for publication. I only ask you to read the few points I have made that I believe deserve attention.
We are grateful for these kind words
Line 43: I agree with this statement, but in my opinion it must also be considered that sometimes the decrease is not so evident and it takes many deliveries to observe a significant variation. In this case, however, the economic damage has already been done as many fertilizations have been done in the meantime. For this reason, preventive screening is useful (This observation is partly explained in the following lines).
This is a good point – we have clarified it using some of the text so eloquently put by the reviewer.
Line 45: The value refers to 2014: is there a more recent data?
We have updated the yearbook reference but the numbers have not changed
Line 60: Would it be possible to insert a reference?
Done (now line 66)
Line 75: In my opinion the identification of translocations involving much larger fragments is however difficult. It depends on what portions of chromosomes are exchanged and what banding profile they have. I would prefer that all references to dimensions be eliminated.
We have removed the figures and reworded accordingly. Now line 78.
Line 106: I suggest to add two lines regarding the G band protocol for karyotypes (considering that there are two figures in the paper)
Done (now line 100)
Line 111: What does this “system” consist of?
We have clarified this. It means the system of looking for XY chimerism (lines 124 and 132)
Line 122: I would also indicate the 95% confidence interval of this percentage (1.33-0.42). In my opinion it is interesting to note that the value reported on line 59 is included in this range.
Done (line 144). See also comment in line 191.
Line 170: I believe that using the described techniques it is not possible to establish whether a translocation is balanced or not. Only the use of other molecular approaches can give this result (as demonstrated in humans: Patsalis et al., Eur J Hum Genet 12:647--653 (2004); Gribble et al., J Med Genet 42:8--16 (2005); De Gregori et al., J Med Genet 44:750--762 (2007); Sismani et al., Mol Cytogenet 1:1--15 (2008).
We take the point, but if the translocation were unbalanced, then the pig would either have never been born, or would have severe congenital abnormalities. We have added a line of clarification (lines 206-210).
Reviewer 2 Report
Dear editor and authors,
I read the paper and I found that the manuscript has merit, but some details in material and methods should be included to let it acceptable. Please be more detailed as possible about the procedures. The word discussion is repeated in the section "5." that should be conclusions. I would expect a more comprehensive discussion.
Thanks for the opportunity.
Author Response
I read the paper and I found that the manuscript has merit, but some details in material and methods should be included to let it acceptable. Please be more detailed as possible about the procedures. The word discussion is repeated in the section "5." that should be conclusions. I would expect a more comprehensive discussion.
We are grateful for these kind words form reviewer 2. A more comprehensive methods and discussion has now been added. The word “conclusions” has been substituted.